# PeerJ

# Cyclic nucleotide binding and structural changes in the isolated GAF domain of *Anabaena* adenylyl cyclase, CyaB2

Kabir Hassan Biswas[1,*], Suguna Badireddy[2], Abinaya Rajendran[1], Ganesh Srinivasan Anand[2] and Sandhya S. Visweswariah[1]

[1] Department of Molecular Reproduction, Development and Genetics, Indian Institute of Science, Bangalore, India
[2] Department of Biological Sciences, National University of Singapore, Singapore, Singapore
[*] Current affiliation: Mechanobiology Institute, National University of Singapore, Singapore, Singapore

## ABSTRACT

GAF domains are a large family of regulatory domains, and a subset are found associated with enzymes involved in cyclic nucleotide (cNMP) metabolism such as adenylyl cyclases and phosphodiesterases. CyaB2, an adenylyl cyclase from *Anabaena*, contains two GAF domains in tandem at the N-terminus and an adenylyl cyclase domain at the C-terminus. Cyclic AMP, but not cGMP, binding to the GAF domains of CyaB2 increases the activity of the cyclase domain leading to enhanced synthesis of cAMP. Here we show that the isolated GAFb domain of CyaB2 can bind both cAMP and cGMP, and enhanced specificity for cAMP is observed only when both the GAFa and the GAFb domains are present in tandem (GAFab domain). *In silico* docking and mutational analysis identified distinct residues important for interaction with either cAMP or cGMP in the GAFb domain. Structural changes associated with ligand binding to the GAF domains could not be detected by bioluminescence resonance energy transfer (BRET) experiments. However, amide hydrogen-deuterium exchange mass spectrometry (HDXMS) experiments provided insights into the structural basis for cAMP-induced allosteric regulation of the GAF domains, and differences in the changes induced by cAMP and cGMP binding to the GAF domain. Thus, our findings could allow the development of molecules that modulate the allosteric regulation by GAF domains present in pharmacologically relevant proteins.

## INTRODUCTION

GAF domains (c**G**MP-specific and -regulated cyclic nucleotide phosphodiesterase, **A**denylyl cyclase, and *E. coli* transcription factor **F**hlA) are a family of protein domains that regulate the function of a variety of domains with which they are associated (*Aravind & Ponting, 1997*; *Charbonneau et al., 1990*). They represent one of the largest families of small molecule-binding regulatory domains, and are found in organisms in all three kingdoms of life (*Anantharaman, Koonin & Aravind, 2001*; *Martinez, Beavo & Hol, 2002*). GAF domains (∼150 amino acids long) are found associated with additional signaling domains such as

Corresponding authors
Kabir Hassan Biswas,
mbikhb@nus.edu.sg
Sandhya S. Visweswariah,
sandhya@mrdg.iisc.ernet.in

the PAS, Sigma54_activat, helix-turn-helix (HTH), PEP_utilizers_C, GGDEF, EAL, HisKA and phosphodiesterase domains (*Aravind & Ponting, 1997*; *Finn et al., 2010*). GAF domains can bind a variety of ligands including tetrapyrroles, formate, haeme, bilin and cyclic nucleotides (*Anantharaman, Koonin & Aravind, 2001*; *Zoraghi, Corbin & Francis, 2004*). Although the sequences of these domains have diverged substantially due to their long evolutionary history (*Aravind et al., 2002*), a motif of five residues (NKFDE) is conserved in most of the characterized cNMP-binding GAF domains (*Zoraghi, Corbin & Francis, 2004*).

The structures of a number of cNMP-binding GAF domains have been solved by X-ray crystallography and NMR. These include the GAF domains in the cGMP-stimulated, cAMP phosphodiesterase, PDE2 [PDB: 1MC0] (*Martinez et al., 2002b*), *Anabaena* CyaB2 adenylyl cyclase [PDB: 1YKD] (*Martinez et al., 2005*) and the cGMP-stimulated, cGMP-specific PDE5 [PDB: 2K31, 3IBJ, 2ZMF, 3FLV] (*Heikaus, Pandit & Klevit, 2009*; *Pandit et al., 2009*; *Russwurm et al., 2011*; *Wang, Robinson & Ke, 2010*). A common structural feature shared by these GAF domain is the presence of six central anti-parallel $\beta$-sheets flanked by $\alpha$-helices on both sides (*Heikaus, Pandit & Klevit, 2009*). The $\beta$-sheets form a curved plane that separates the $\alpha$-helices into two groups. The curved plane of the antiparallel $\beta$-sheets serves as the base of the ligand-binding pocket, and the rest of the ligand-binding pocket is covered by helices $\alpha3$, $\alpha4$, and some loop regions. Helices $\alpha2$ and $\alpha5$ are present on the opposite side of the ligand-binding pocket. In CyaB2, helix $\alpha2$ connects the GAFb domain to N-terminal GAFa domain in CyaB2 and helix $\alpha5$ connects the GAF domain to the C-terminal PAS and adenylyl cyclase effector domains. Cyclic AMP is buried within the ligand-binding pocket (Fig. 1A), and important residues in the ligand binding pocket that interact with cAMP include Arg 291 (H-bond with N1 of cAMP), Thr 293 (H-bond with N6 and N7 of cAMP), Asp 356 and Asn 359 (water mediated H-bond with N3 of cAMP), and Ile 308 (hydrophobic contact to the adenine ring of cAMP).

Most of the cNMP binding GAF domains show high specificity towards a specific cyclic nucleotide, but the basis for this selectivity in some GAF domains still remains unknown. Substitution of a region of the CyaB1 GAFb domain with that of a corresponding region in the cGMP-binding GAF domain of PDE2, allowed CyaB1 to show cGMP-enhanced adenylyl cyclase activity. However, the converse experiment in which amino acids in the PDE2 GAF domain were replaced with those from CyaB1 did not lead to altered specificity (*Linder et al., 2007*). In addition, structural studies combined with mutational analysis of the GAFa domain of PDE5 suggested that a key residue (Asp 164) allowed the discrimination between cAMP and cGMP (*Heikaus et al., 2008*).

In the present study, we show by direct ligand binding assays that the specificity of nucleotide binding is reduced in an isolated GAF domain, as compared to the tandem GAFab domains of CyaB2. *In silico* docking and mutation of key interacting residues provided insights on cGMP binding, and HDXMS identified diverse structural changes induced by cAMP and cGMP.

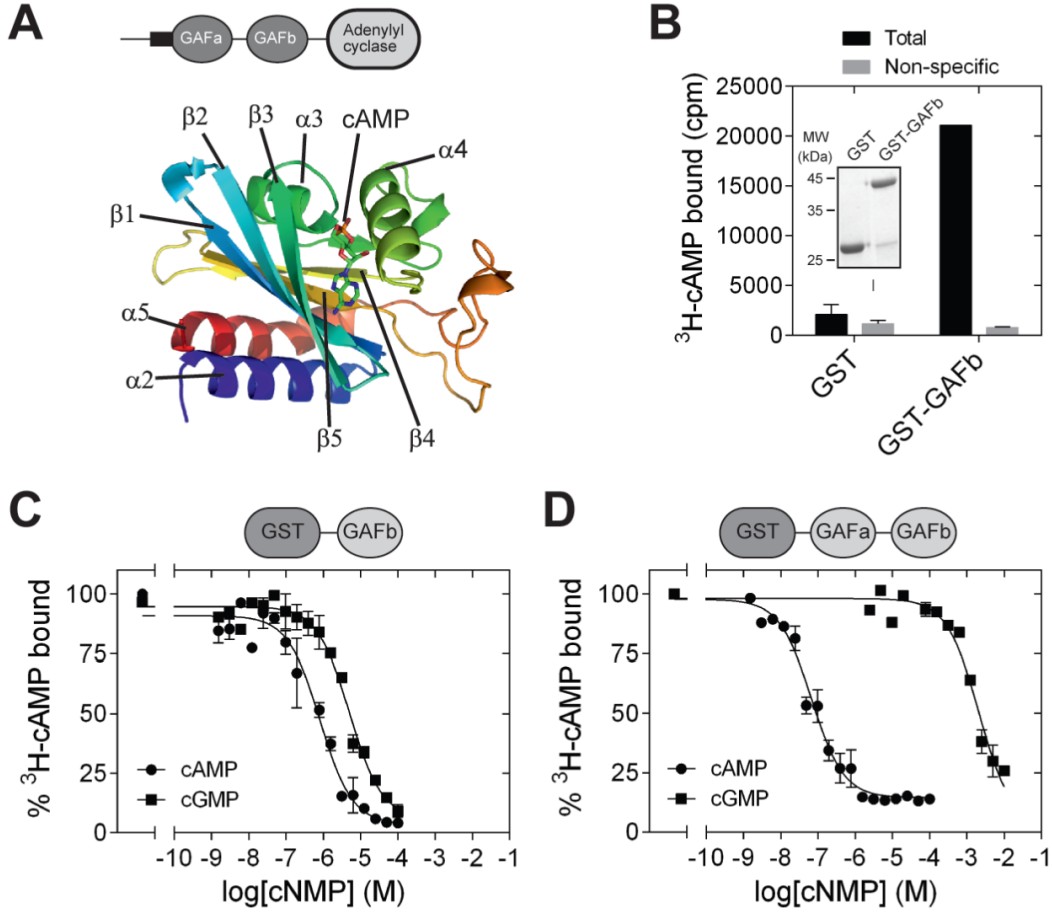

**Figure 1 Isolated GAFb domain binds both cAMP and cGMP.** (A) Cartoon representation of the structure of the GAFb domain illustrating various secondary structure elements and bound cAMP molecule [PDB: 1YKD (*Martinez et al., 2005*)]. (B) Proteins (∼1 µg) bound to glutathione beads were incubated with $^3$H-cAMP (∼1 nM) in the absence or presence of 10 µM unlabeled cAMP. Data shown is a representative of assays performed thrice in duplicates, and values shown are mean ±S. E. M. The inset shows a Coomassie stained gel picture of the purified proteins used in the assay. (C) and (D) Purified GST-GAFb (C) or GST-GAFab (D) proteins bound to beads were incubated with ∼1 nM $^3$H-cAMP and increasing concentration of unlabeled cAMP or cGMP. Data shown is mean ±S. E. M of duplicate determinations and is representative of independent assays.

## MATERIALS AND METHODS

### Generation of various GAF domain constructs and mutagenesis

The nucleotide sequence of the GAFb domain of CyaB2 from *Anabaena* sp. PCC 7120 spanning residues L270 to L431 was amplified by PCR from the full-length CyaB2 gene cloned into pQE30 plasmid (pQE30-CyaB2 (*Bruder et al., 2005*)) using primers GAFbf793 (5′ CTGGGATCCGGTACCCTGGATTTAGAAGATACCC 3′) and GAFbr1279 (5′ ACACTCGAGCGATATCTAAAGCCACCCCGGC 3′). The PCR product was directly cloned into pGEM-T-Easy vector (Promega, Southampton, UK) to generate the plasmid pGEM-T-GAFb and insert was sequenced. To generate a GST fusion protein for cyclic nucleotide binding experiments and His$_6$-tagged protein for HDXMS experiments, the

GAFb nucleotide sequence was released and subcloned into pGEX-6p-1 plasmid vector (GE Healthcare, Little Chalfont, UK) and pPRO-Ex-B plasmid vector (Invitrogen, Lofer, Austria), respectively, using *Bam*HI and *Xho*I sites, resulting in the pGEX-6p-1-GAFb and pPRO-Ex-GAFb plasmid. To generate a BRET-based sensor, the GAFb gene fragment was released and subcloned into the pGFP$^2$-MCS-Rluc plasmid vector (PerkinElmer Life Sciences, Waltham, Massachusetts, USA) using *Kpn*I and *Eco*RV sites, resulting in the pGFP$^2$-GAFb-Rluc plasmid. The I308A and T293A mutations in the GAFb domain was introduced using a single primer (*Shenoy & Visweswariah, 2003*) and primers GAFb_CyaB2(I308A) (5′ GGACGAAAGCTACCCAAGATAATGGTTCTACTAAGG 3′) and GAFb_CyaB2(T293A). (5′ GATGAACGCGGACCGCAGTGCCTTATGGCTGATAG 3′). Mutations were confirmed by sequencing.

The nucleotide sequence of the tandem GAFab domains of CyaB2 spanning residues S77 to L431 was amplified by PCR from the full-length CyaB2 gene cloned into pQE30 plasmid (pQE30-CyaB2 *Bruder et al., 2005*) using primers GAFaf216 (5′ GTCAATGTTGGGATCC-CACGGTACCGAAAATATCCTGC 3′) and GAFbr1279. The PCR product was directly cloned into pGEM-T-Easy vector (Promega, Southampton, UK) and sequenced. To express a GST fusion protein, the tandem GAFab domain gene fragment was subcloned into pGEX-6p-1 plasmid vector (GE Healthcare, Little Chalfont, UK) using *Bam*HI and *Xho*I sites, resulting in the pGEX-6p-1-GAFab plasmid. To generate a BRET-based sensor construct, the tandem GAFab domain gene fragment was subcloned into the pGFP$^2$-MCS-Rluc plasmid vector (PerkinElmer Life Sciences, Waltham, Massachusetts, USA) using *Eco*RV and *Xho*I sites, resulting in the generation of pGFP$^2$-GAFab-Rluc plasmid.

The nucleotide sequence encoding residues M1 to L431 was PCR amplified from the full-length CyaB2 gene cloned into pQE30 plasmid (pQE30-CyaB2 *Bruder et al., 2005*) using primers GAFab_CyaB2f1 (5′ ATATGGATCCGGTACCATGTCATTGCAACAGCG 3′) and GAFb_CyaB2r1271 (5′ ACACTCGAGCGATATCTAAAGCCACCCCGGC 3′) and subcloned into pGEX-6p-1 plasmid vector (GE Healthcare, Little Chalfont, UK) using *Bam*HI and *Xho*I sites, resulting in the pGEX-6p-1-NterGAFab plasmid and sequenced. To generate a BRET-based sensor construct containing N-terminal regional along with tandem GAFab domain (NterGAFab), gene fragment encoding NterGAFab domain was subcloned into the pGFP$^2$-MCS-Rluc plasmid vector (PerkinElmer Life Sciences, Waltham, Massachusetts, USA) using *Eco*RV and *Xho*I sites, resulting in the generation of pGFP$^2$-NterGAFab-Rluc plasmid.

## Expression and purification of the GAF domain constructs of CyaB2

To express and purify GST fusion proteins, *E. coli* BL21 (DE3) cells were transformed with specific plasmid and cells were induced using 100 μM IPTG at 37 °C for 3 h. Cells were collected by centrifugation and cell pellet was resuspended in lysis buffer containing 50 mM Tris (pH 8.2 at 4 °C), 100 mM NaCl, 10% glycerol, 2 mM PMSF, 1 mM benzamidine. Cells were lysed by sonication and lysate was centrifuged at 30,000 g

for 30 min at 4 °C. Supernatant was collected and interacted with pre-equilibrated Glutathione Sepharose 4B beads (GE Healthcare Life Sciences, Little Chalfont, UK) at 4 °C for 1 h. Post interaction, beads were washed with buffer containing 50 mM Tris (pH 8.2 at 4 °C), 100 mM NaCl, 0.1% TritonX-100 followed by three washes with buffer containing 50 mM Tris (pH 8.2 at 4 °C), 100 mM NaCl, 10% glycerol. The protein-bound GSH beads were resuspended in buffer containing 25 mM HEPES, 100 mM NaCl and 10% glycerol and stored at 4 °C till further use.

To express the $His_6$-GAFb protein, *E. coli* BL21DE3 cyc$^-$ cells were transformed with pPRO-Ex-B-GAFb plasmid DNA and induced with 500 µM IPTG for 3 h at 37 °C (*Nambi, Basu & Visweswariah, 2010*). Cells were harvested by centrifugation at 6,000 g for 20 min and the cell pellet was resuspended in lysis buffer [20 mM Tris–HCl (pH 7.5), 100 mM NaCl, 5 mM $\beta$-mercaptoethanol, 5 mM imidazole and EDTA free protease inhibitor tablet (Roche, Basel, Switzerland)]. Cells were lysed by sonication and centrifuged at 17,000 g at 4 °C for 40 min. The supernatant was collected and incubated with Talon resin (Invitrogen, Lofer, Austria) at 4 °C for 1 h. The resin was then transferred into columns, and washed with lysis buffer and wash buffer (lysis buffer with 10 mM imidazole) followed by elution buffer containing lysis buffer with 150 mM imidazole. Further purification was achieved by size-exclusion chromatography on a Superdex 200 column using the AKTA FPLC System (GE Healthcare Life Sciences, Little Chalfont, UK).

## Cyclic nucleotide binding assays

Cyclic nucleotide binding assays were carried out essentially as described earlier (*Sopory et al., 2003*) using 1–5 µg of purified GST fusion proteins bound to glutathione beads in buffer containing 25 mM HEPES, 100 mM NaCl, 10% glycerol and 200 µM PMSF, in the presence of 2,8 [$^3$H]-cAMP ($\sim$100,000 dpm; 28.1 Ci/mmol; MPI Biomedicals, Mattawan, Michigan, USA) either in the absence or presence of unlabeled cAMP or cGMP, in a total reaction volume of 50 µL. Reactions were incubated at 37 °C for 1 h and then filtered through GF/C filters (Whatman, Maidstone, UK), which were then washed with 6 mL of ice-cold washing buffer (10 mM Tris, pH 7.5, 100 mM NaCl and 10% glycerol). The filters were then dried and radioactivity was measured in a liquid scintillation counter.

## Docking of cyclic nucleotides to the GAFb domain of CyaB2

Docking was performed using AutoDock (Version 3.0.5) (*Morris et al., 1998*) implemented using AutoDock Tools (Molecular Graphics Laboratory, La Jolla, California, USA). The performance of AutoDock was tested by first docking cAMP into the GAFb domain. For docking cNMPs to the GAFb domain of CyaB2, the atomic structure comprising residues Leu270 to Leu431 of CyaB2 (PDB: 1YKD) was selected to be used as the macromolecule (*Martinez et al., 2005*). All water molecules and cAMP were removed from the structure before docking. Atomic coordinates of cAMP and cGMP were generated from the SMILES structure descriptor format, available in the PubChem database, using Online SMILES Translator and Structure File Generator (http://cactus.nci.nih.gov/services/translate/). Charges were added to the atoms and the final structure file for AutoDock was prepared using the Dundee PRODRG2 Server (http://davapc1.bioch.dundee.ac.uk/programs/

prodrg/) (*Schuttelkopf & van Aalten, 2004*). Atomic volume and solvation parameters were assigned to the protein molecule using default values. Polar hydrogen atoms were added and Kollman charges were assigned using the built in function in AutoDock Tools. Grids were used at a spacing of 0.375 Å covering the cAMP binding site in a cube of $90 \times 90 \times 90$ grid points with the grid center placed near the phosphate group of cAMP as found in the crystal structure. Docking was performed with a Lamarckian genetic algorithm with a total of 50 genetic algorithm runs for cAMP and 20 for cGMP. Other docking parameters were: population size = 50, mutation rate = 0.02, crossover rate = 0.8, number of genetic algorithm evaluations = 250,000, number of genetic algorithm generations = 2,700,000 and genetic algorithm elitism = 1. Results were analyzed using command get_docked and Pymol (The PyMOL Molecular Graphics System, Version 1.5.0.4 Schrödinger, LLC).

## Cell culture and transfection

Human embryonic kidney (HEK) 293T cells were maintained in Dulbecco's modified Eagle's media (DMEM) with 10% fetal calf serum, 120 mg/L penicillin and 270 mg/L streptomycin at 37 °C in a 5% $CO_2$ humidified incubator. Transfection was performed with polyethyleneimine lipid according to manufacturers' protocols. Expression of proteins was monitored by Western Blot analysis using an antibody raised in rabbit against the Rluc protein and described below.

## Generation of polyclonal antibody against Rluc

Polyclonal antibodies against Rluc protein was raised in rabbits using $His_6$-Rluc protein essentially as described previously (*Bakre, Sopory & Visweswariah, 2000*). Rluc gene was released from pRluc-N1 plasmid vector (PerkinElmer Life Sciences, Waltham, Massachusetts, USA) and subcloned into pPRO-Ex-C (Invitrogen, Lofer, Austria) using *Bam*HI and *Xba*I sites to generate the pPRO-Ex-C-Rluc plasmid. To express the protein, *E. coli* BL21DE3 were transformed with pPRO-Ex-C-Rluc plasmid and induced with 100 μM IPTG. $His_6$-Rluc protein formed inclusion bodies. Aggregated protein was solubilized using urea and used for antibody generation. The primary dose of immunogen (∼500 μg) was in Freund's complete adjuvant and booster dose of immunogen (∼250 μg) was in Freund's incomplete adjuvant. The presence of antibody was detected by ELISA and Western Blot analysis.

## *In vitro* BRET assays

All BRET assays were performed using the $BRET^2$ assay components i.e., acceptor—$GFP^2$, donor—Rluc and Rluc substrate—Coelenterazine 400a. In vitro BRET assays were performed as described previously (*Biswas, Sopory & Visweswariah, 2008*; *Biswas & Visweswariah, 2011*). HEK 293T cells transfected with appropriate plasmids were lysed in a buffer of 50 mM HEPES (pH 7.5), containing 2 mM EDTA, 1 mM dithiothreitol, 100 mM NaCl, 10 mM sodium pyrophosphate, 80 μM $\beta$-glycerophosphate, 1 mM benzamidine, 1 μg/mL aprotinin, 1 μg/mL leupeptin, 5 μg/mL soybean trypsin inhibitor, 100 μM sodium orthovanadate and 10 % glycerol. Following brief sonication, the lysates was centrifuged at 13,000 g and the cytosol was collected. Aliquots of the cytosol were incubated with

Peer**J**

1 mM cNMP in buffer of 50 mM HEPES, pH 7.5, containing 100 mM NaCl in a total volume of 40 μL, at 37 °C for 10 min. Coelenterazine 400a (Molecular Imaging Products, Bend, Oregon, USA) was added to a final concentration of 5 μM and emissions were collected for 0.8 s in a Victor[3] microplate reader (PerkinElmer Life Sciences, Waltham, Massachusetts, USA). Emission filters used for Rluc and GFP[2] emission were 410 nm (bandpass 80 nm) and 515 nm (bandpass 30 nm), respectively. BRET was calculated as the ratio of GFP emission per second to Rluc emission per second, and the average of three such measurements is reported.

## Cellular BRET assays

HEK 293T cells were transfected with pGFP[2]-GAFb-Rluc plasmid in 10 cm tissue culture dishes. Forty eight hours post transfection, medium was removed, and monolayers treated with Dulbecco's phosphate buffered saline containing 5 mM EDTA for 5 min at 37 °C in the incubator following which the EDTA solution was removed, and cells resuspended in phenol-red free DMEM, containing 10% fetal calf serum. Cells ($\sim$10$^5$) were then treated with 100 μM of either forskolin (for 5 min) or sodium nitroprusside (for 2 min). BRET was determined as described above.

## Intracellular cNMP estimation

Intracellular levels of cyclic nucleotide monophosphates (cNMP) were measured from the cells used for BRET measurements, or parallely transfected and treated cells. Cells were lysed in 0.1 N HCl and cNMP was measured by radioimmunoassay as described earlier (*Bakre, Ghanekar & Visweswariah, 2000*).

## HDXMS of the GAFb domain of CyaB2

The cAMP-free GAFb domain purified by size exclusion chromatography was concentrated to 50 μM using vivaspin concentrators (Sartorius Stedim Biotech GmbH, Goettingen, Germany). Samples were prepared by adding either 1 mM cAMP or cGMP to apo GAFb domain protein. 2 μL each of apo, cAMP-, or cGMP-bound GAFb domain in 20 mM Tris–HCl (pH 7.5), 100 mM NaCl and 5 mM beta-mercaptoethanol buffer were diluted and incubated with 18 μL of D$_2$O (99.90%) (Fluka BioChemika, Buchs, Switzerland) resulting in a final deuterium concentration of 90%. Hydrogen-deuterium exchange was carried out at 20 °C for various time points (0.5, 1, 2, 5 and 10 min). The exchange reaction was quenched by adding 40 μl of prechilled quench buffer (0.1% trifluoroacetic acid (Fluka BioChemika, Buchs, Switzerland) to get a final pH read of 2.5. An aliquot (50 μl) of the quenched sample was then injected on to a chilled nanoUPLC sample manager (beta test version, Waters, Milford, Massachusetts, USA) as previously described (*Badireddy et al., 2011*; *Wales et al., 2008*). Peptides were detected and sequenced and mass was measured on a Synapt HDMS mass spectrometer (Synapt, Waters, Manchester, UK) acquiring in the MS$^E$ mode, a nonbiased, nonselective CID method (*Bateman et al., 2002*; *Li et al., 2009*; *Shen et al., 2009*; *Silva et al., 2005*).

Sequence identifications were made from MS$^E$ data from undeuterated samples using ProteinLynx Global Server 2.4 (beta test version; Waters, Milford, Massachusetts, USA)

(*Geromanos et al., 2009*; *Li et al., 2009*) and searched against sequence of GAFb domain with no enzyme specified and no modifications of amino acids. Identifications were only considered if they appeared at least twice out of three replicate runs. The precursor ion mass tolerance was set at <10 ppm and fragment ion tolerance was set at <20 ppm. Only those peptides that satisfied the above criteria through Database search pass 1 were selected and are listed in Table 1 (*Li et al., 2009*). The default criterion for false positive identification (Value = 4) was applied. These results showed that $MS^E$ data searched with PLGS 2.4 maximized identification of peptides and were used for deuterium exchange analysis. These identifications were mapped to subsequent deuteration experiments using prototype custom software (HDX browser, Waters, Milford, Massachusetts, USA). Data on each individual peptide at all periods were extracted using this software, and exported to HX-Express (*Weis, Engen & Kass, 2006*) for analysis. A total number of 38 peptide fragments yielded primary sequence coverage of 96%. Continuous instrument calibration was carried out with Glu-fibrinogen peptide at 100 fmol/μl. We also visually analyzed the data to ensure only well resolved peptide isotopic envelopes were subjected to quantitative analysis.

### Statistical analysis

All experimental data were analyzed using GraphPad Prism and represent the mean ± S.E.M.

## RESULTS

### GAFb domain of CyaB2 binds both cAMP and cGMP

GAF domains associated with enzymes such as nucleotide cyclases and phosphodiesterases are often present in tandem repeats (*Bruder, Schultz & Schultz, 2006*; *Schultz, 2009*). In the case of CyaB2, both the GAFa and GAFb domains bind cAMP. However, binding of cAMP to the GAFb domain is likely to trigger the conformational changes in the protein that enhance adenylyl cyclase activity. We therefore tested if an isolated GAFb domain of CyaB2 was able to bind ligand by direct radiolabeled cyclic nucleotide binding assays. The isolated GAFb domain encompassing residues Leu 270 to Leu 431 fused to GST was expressed in bacteria and purified. High affinity cAMP binding, with a $K_D$ of 0.8 ± 0.2 μM (Figs. 1B and 1C) was observed, and was similar to the $EC_{50}$ value (∼1.3 μM) reported previously from assays monitoring cAMP-mediated activation of a related adenylyl cyclase domain (*Bruder et al., 2005*). This result along with our previous studies using the isolated GAFa domain of PDE5 (*Biswas, Sopory & Visweswariah, 2008*) establishes that isolated GAF domains are able to bind their respective ligand even in the absence of the second GAF domain, or other associated catalytic domains.

Previous studies have shown that the GAF domains associated with nucleotide cyclases and phosphodiesterases show specificity in binding either cAMP or GMP. For instance, the GAFa domain of PDE5 specifically binds cGMP (*Biswas, Sopory & Visweswariah, 2008*; *Sopory et al., 2003*), while the GAFb domain of PDE2 binds cAMP (*Martinez et al., 2002b*). The tandem GAF domains of CyaB2 have also been shown to be highly specific for cAMP

as monitored by the activation of the CyaB1-CyaB2 fusion protein (*Bruder et al., 2005*). However, we observed that cGMP could efficiently compete with cAMP for binding to the isolated GAFb domain, with an $IC_{50}$ of $7.6 \pm 1.9\,\mu M$ (Fig. 1C), in contrast to the specificity of nucleotide-mediated activation of the adenylyl cyclase domain fused to the tandem GAF domains of CyaB2 (*Bruder et al., 2005*).

To make a direct comparison of binding specificity, we performed cyclic nucleotide binding assays with a construct containing both the GAF domains. For this, a GST fusion of the tandem GAFab domains encompassing residues S77 to L431 was expressed and purified. Competition radiolabeled nucleotide binding assays with cAMP and cGMP revealed that the tandem GAFab domain show much higher affinity for cAMP ($IC_{50}$ $0.05 \pm 0.02\,\mu M$; 16-fold higher affinity than the isolated GAFb domain) and a much reduced affinity for cGMP ($IC_{50}$ $2,651 \pm 870\,\mu M$; 348-fold lower affinity compared to the isolated GAFb domain) (Fig. 1D). Therefore, while the isolated GAFb domain showed only 10-fold selectivity towards cAMP, the tandem GAFab domain showed much higher ($\sim$50, 000-fold) selectivity for cAMP. Thus, although nucleotide binding is preserved in the isolated GAFb domain, removal of the associated GAFa domain results in a reduction in both affinity and specificity of nucleotide binding.

### In silico docking of cGMP to the GAFb domain of CyaB2

To gain insight into the mechanism by which cGMP could interact with the isolated GAFb domain, we performed *in silico* docking of cGMP on the structure of the GAFb domain. We removed the bound cAMP molecule from the crystal structure of the GAFb domain and used it as the receptor (*Martinez et al., 2005*), and all dockings were performed using AutoDock (*Morris et al., 1998*). We first tested the performance of the *in silico* experiment by docking cAMP and comparing the results with the original crystal structure data. Indeed, cAMP could be docked into the GAFb domain in a pose closely mimicking the original structure, with an RMSD of $\sim$0.2 Å between the docked and crystallized cAMP molecule. Following this, we performed a blind docking of cGMP molecule with 20 independent docking runs that resulted in the same number of final docked conformers. All cGMP conformers except one were docked into the cNMP binding pocket of the GAFb domain (Fig. 2B). Interestingly, the majority of conformers were found to interact with the GAFb domain in an orientation that was distinct from cAMP (designated as mode 1) while only two conformers were found to interact in a mode that was similar to crystal structure bound cAMP (designated as mode 2) (Figs. 2A and 2B). Further, the energy of interaction was lower for mode 1 conformers compared to mode 2 conformers. Detailed analysis of a representative mode 1 conformer revealed that this type of conformation could be stabilized by H-bond interactions of the O6 of cGMP with the side chain of Asn 359, and N7 with Asp 356 present in the helix $\alpha$4. On the other hand, mode 2 conformer could be stabilized by H-bond interactions between N7 and O6, and Thr 293 in addition to hydrophobic interaction with I308 (Fig. 2A).

We then performed mutational analysis to get further insights into ligand binding to the GAFb domain and validate the docking results. Analysis of the crystal structure of the
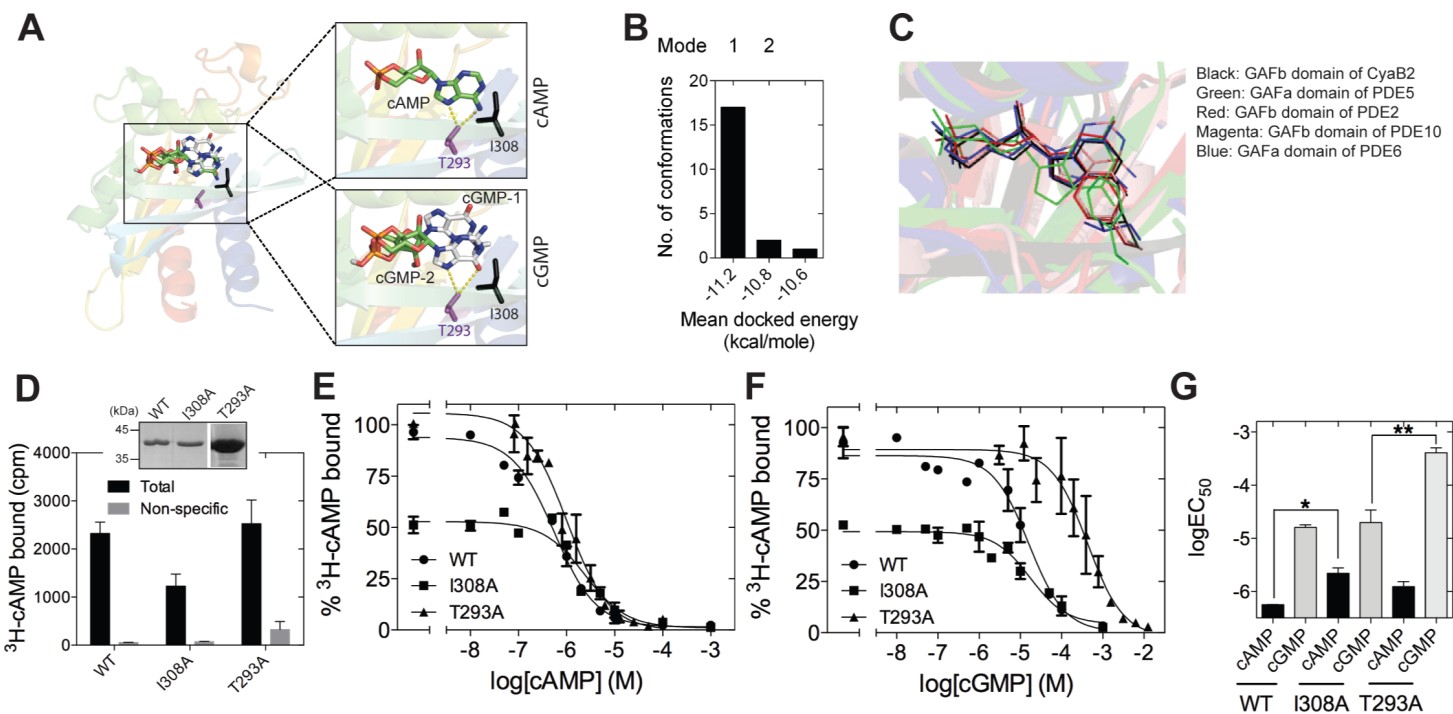

**Figure 2 Ligand binding to the isolated GAFb domain of CyaB2.** (A) Cartoon representation of the GAFb domain (PDB: 1YKD *Martinez et al., 2005*) with docked cAMP and cGMP conformers. Insets show zoomed in view of the ligand binding pocket of GAFb with docked cAMP and two different clusters of docked cGMP conformers indicated as cGMP-1 and cGMP-2. Side chains of T293 and I308 are also shown along with their interaction with the docked ligands. (B) Distribution of docked cGMP conformers obtained clustering with an RMSD of 0.5 Å. Mode 1 represents the cluster with maximum number of conformers and with lowest energy while mode 2 represents the cluster with cGMP docked in a way that mimics bound cAMP. Third cluster consisting of one conformer out of 20 was docked outside the ligand-binding pocket and therefore, has not been shown. (C) Conserved interaction between high affinity ligand, either cAMP or cGMP, with the residue equivalent to I308 in different cyclic nucleotide binding GAF domains (GAFb domain of CyaB2–cAMP [PDB: 1YKD]; GAFb domain of PDE2–cGMP [PDB: 1MCO]; GAFa domain of PDE5–cGMP [PDB: 2K31]; GAFb domain of PDE10–cAMP [PDB: 2E4S]; GAFa domain of PDE6–cGMP [PDB: 3DBA]. (D) Wild type (WT), I308A or T293A mutant GST-fusion proteins (~1–5 µg) bound to glutathione beads were incubated with $^3$H-cAMP (~1 nM) in the absence or presence of 10 µM unlabeled cAMP. Data shown is a representative of assays performed at least twice in duplicate, and values shown are mean ± S. E. M. for 1 µg protein. The inset shows a Coomassie stained gel picture of the purified proteins used in the assay. (E) and (F) Purified wild type, I308A or T293A mutant GST fusion proteins bound to beads were incubated with ~1 nM $^3$H-cAMP and increasing concentration of unlabeled cAMP (E) or cGMP (F). Data shown is mean ±S. E. M of duplicate determinations and is representative of independent assays. (G) Log values of the $IC_{50}$ obtained for the wild type, I308A or T293A mutant proteins with either cAMP or cGMP are plotted. Data shown is mean ± S. E. M. of $IC_{50}$ values determined from multiple independent assays.

GAFb domain showed an interaction between the side chain of I308 and the adenosine ring of cAMP (*Martinez et al., 2005*). An equivalent interaction is conserved in cyclic nucleotide binding GAF domains (Fig. 2C) but this interaction should be dispensable if cGMP bound as predicted in mode 1. However, interaction of cGMP with Thr 293 should be important for binding in mode 1 and could be dispensable for binding in mode 2. Therefore, we mutated the Ile 308 and Thr 293 to A (I308A & T293A mutants), and performed radiolabeled ligand binding assays with the purified mutant GAFb domains. We observed an expected ~50% reduction in the binding of $^3$H-cAMP to the I308A mutant

GAFb domain (Fig. 2D), correlated with a significant reduction in the affinity for cAMP (Figs. 2E and 2G). In contrast, the affinity for cGMP remained unaltered in this mutant protein (Figs. 2F and 2G). The T293A mutant protein, however, showed a reduction in the affinity for cGMP (Figs. 2E and 2G) but no change in the affinity of cAMP binding (Figs. 2F and 2G). Taken together, these results suggest that cGMP binds to the GAFb domain as seen in mode 2. Additionally, unlike cAMP, cGMP binding does not require interaction with Ile 308, indicating subtle differences in residues interacting with the nucleotides.

## Ligand induced structural changes in the GAF domain are subtle and cannot be detected by BRET

Ligand binding to the GAF domains in CyaB2 is highly cooperative (*Bruder et al., 2005*) and acts as an allosteric signal that results in the activation of the C-terminal adenylyl cyclase domain (*Kanacher et al., 2002*). This implies that ligand binding to the GAFb domain may result in a structural change that is communicated to both the N-terminal GAFa domain and the C-terminal adenylyl cyclase domain (Fig. 1A). We utilized BRET technology to determine if ligand binding to the GAFb domain alone results in significant structural rearrangements. We have earlier used this strategy successfully to detect cGMP-induced structural changes in the isolated GAFa domain of PDE5 (*Biswas, Sopory & Visweswariah, 2008*), as well as in the full-length PDE5 (*Biswas & Visweswariah, 2011*).

We generated a fusion protein containing GFP[2] at the N- and Rluc protein to the C-terminal ends of GAFb (GAFb sensor) (Fig. 3A). The GAFb sensor protein was expressed in HEK293T cells and detected by western blot analysis using antibodies to Rluc (Fig. 3B). Lysates prepared from HEK293T cells expressing the GAFb sensor were incubated in the absence or presence of 1 mM cAMP or cGMP, and BRET was measured. We used the F163A mutant GAFa domain of PDE5, which binds both cAMP and cGMP, for the purpose of comparison (*Biswas, Sopory & Visweswariah, 2008*). Basal BRET ratio of the GAFb sensor was higher compared to the PDE5 GAFa(F163A) sensor (Fig. 3C), indicating that the GAFb sensor expressed in mammalian cells is folded and could potentially bind ligand. Importantly, unlike the PDE5 GAFa(F163A) sensor which showed a large increase in the BRET ratio in the presence of both cAMP and cGMP, no change in the BRET ratio was observed for the GAFb sensor in the presence of either cAMP or cGMP (Fig. 3C). To rule out the requirement of any cellular factor for the induction of structural changes in the GAF domain, we performed experiments with live cells expressing the GAFb sensor. Intracellular levels of cAMP were elevated using forskolin, (*Litosch et al., 1982*), and intracellular cGMP levels were increased by treating cells with sodium nitroprusside (SNP) (*Murad, 1986*). Although both forskolin and SNP treatment resulted in increased intracellular levels of cAMP and cGMP, respectively, no significant changes were observed in the BRET ratios (Figs. 3D and 3E).

The lack of change in BRET may indicate that either there is no substantial structural change induced in the GAFb domain on ligand binding, or the change in conformation induced by ligand binding could not be detected due to lack of a specific structural element in the construct used for BRET (*Russwurm et al., 2007*). We therefore generated fusion

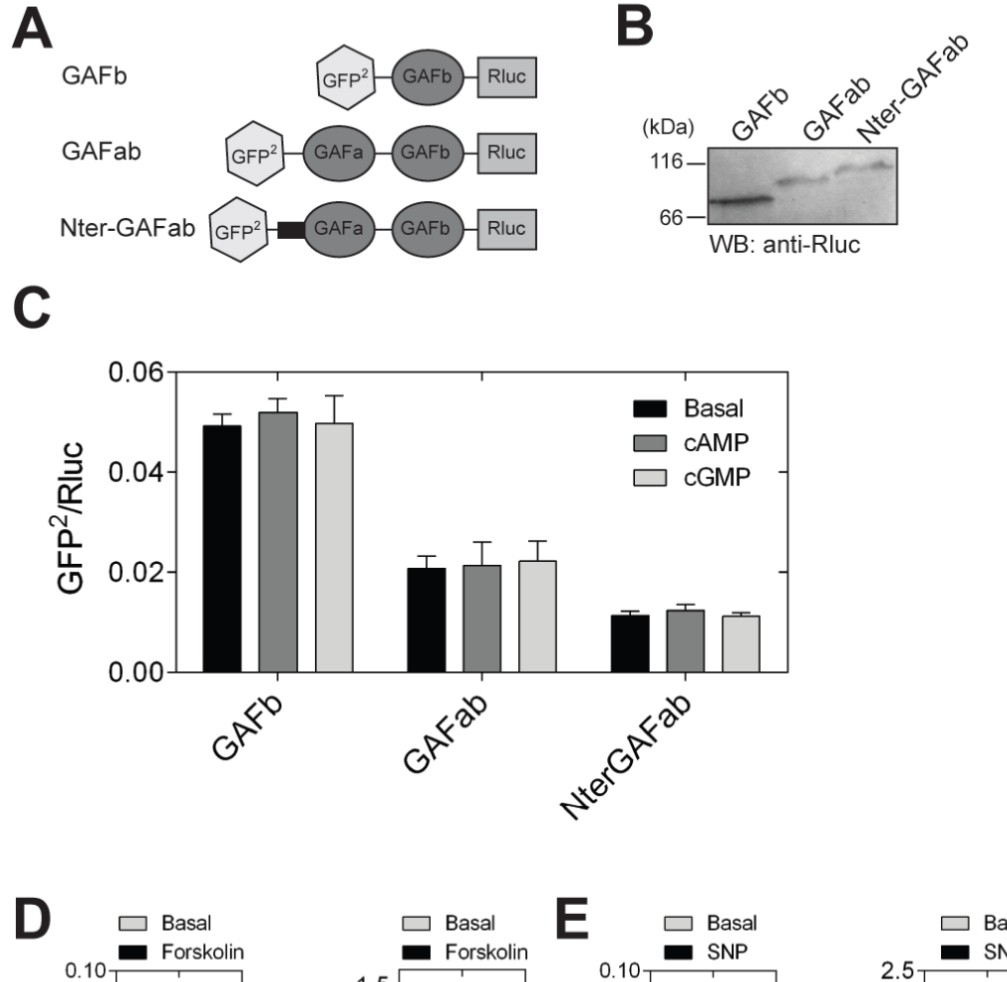

**Figure 3 Ligand binding to the GAFb domain does not result in structural changes at the N- and C-termini.** (A) Diagrammatic representation of various BRET-based sensor constructs used in the study. [**B**] Western blot analysis using anti-Rluc polyclonal antibody to confirm the expression of GAFb, GAFab and NterGAFab sensor constructs. Expected molecular weight of the GAFb, GAFab and NterGAFab sensor constructs are 82.7, 103.8 and 112.7 kDa, respectively. (C) Lysates prepared from cells expressing the PDE5 GAFa(F163A), GAFb, GAFab and NterGAFab sensor constructs were incubated in the absence or presence of 1 mM cAMP or cGMP at 37 °C for 10 min followed by BRET measurement. (D) and (E) HEK293T cells transfected with the GAFb sensor were treated with 100 μM of either forskolin (5 min) (D) or SNP (2 min) (E) at 37 °C and BRET was determined. Intracellular levels of cAMP or cGMP were determined using a parallel set of cells. Data shown are mean ±S.E.M from a representative experiment performed in triplicate.

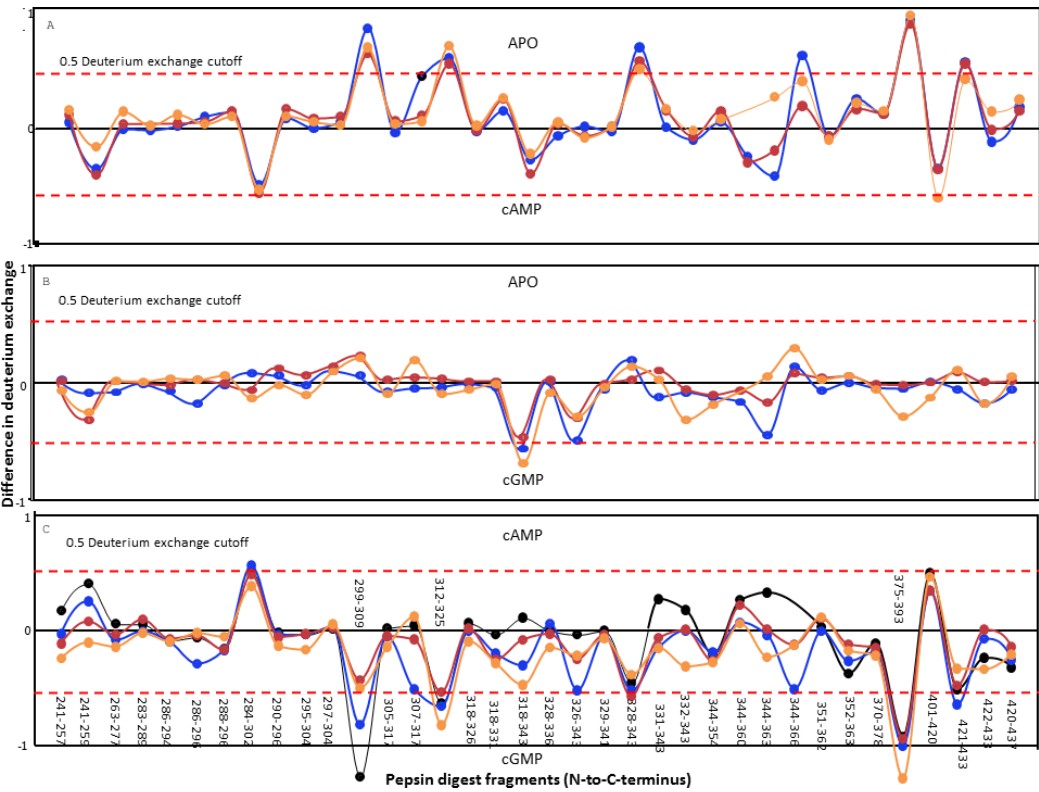

**Figure 4 Protein-wide overview of structural changes induced in the GAFb domain of CyaB2 on ligand binding.** Absolute difference in numbers of deuterons (inferred from difference in mass in Daltons (Da) ($y$ axis) between (A) free and cAMP-bound, (B) free and cGMP-bound and (C) cAMP- and cGMP-bound (bottom) states is plotted for each pepsin digest fragment listed from the N- to C-terminus ($x$ axis) of GAFb domain for each deuterium exchange time point in a difference plot. Time = 0.5 min (orange), 2 min (red), 5 min (blue) and 10 min (black). Shifts in the positive scale represent increases in deuterium exchange and shifts in the negative scale represent decreases in deuterium exchange. A difference of 0.5 Da is considered significant (dashed red line). Plots were generated using the software DYNAMX (Ver. 2.0 Waters). Each point represents a pepsin digest fragment.

constructs containing either the tandem GAFab domains or the tandem GAFab domains along with the complete N-terminal region of CyaB2 (called as GAFab and NterGAFab sensors, respectively). Expression levels of these proteins were lower than that of the isolated GAFb domain (Fig. 3B), and the basal BRET of the constructs decreased in the order GAFb > GAFab > NterGAFab (Fig. 3C). This change in the basal BRET ratio suggested that we were able to detect spatial positioning of GFP$^2$ and Rluc in the sensor constructs. However, incubation of lysates prepared from cells expressing either the GAFab or the NterGAFab sensor with cAMP or cGMP (1 mM) did not result in a significant alteration in the BRET (Fig. 3C).

## Distinct changes in the dynamics of the GAFb domain of CyaB2 induced by cAMP and cGMP binding

The lack of observable change in the BRET of the GAF domains of CyaB2 on ligand binding was intriguing. We therefore decided to monitor more subtle structural changes

in the GAFb domain at a higher resolution using amide hydrogen/deuterium exchange mass spectrometry (HDXMS). His$_6$-tagged GAFb domain was expressed in bacteria and purified for use in these experiments. Complete pepsin digestion of the protein under deuterium exchange quench conditions (pH = 2.5) resulted in the generation of multiple peptide fragments across the domain with >90% sequence coverage (Fig. 4), thus providing a detailed overview of the solvent accessibility and dynamics of the GAFb domain at peptide resolution.

A comparison of amide exchange of various peptides in the absence and presence of cAMP showed a decrease in exchange at the central core region comprising the ligand-binding pocket (Fig. 4A), suggesting a 'closing' of the 'open' ligand binding pocket of the GAFb domain. These included peptides (299–317) spanning parts of $\beta$1 and $\beta$2 sheets, and $\beta$1–$\beta$2 loop showed lower dynamics in the presence of cAMP. The peptide (299–309) contains two residues that interact with cAMP, namely Ile 308 and Thr 309. As discussed previously, Ile 308 provides hydrophobic stacking interaction to cAMP while Thr 309 interacts with the N6 of cAMP, forming H-bond through the peptide backbone carbonyl oxygen.

Interestingly, cAMP binding resulted in an increased solvent accessibility of peptides arising from the N- and C-terminal helices. The N-terminal $\alpha$2 helix connects the GAFb domain to the GAFa domain, and the C-terminal $\alpha$5 helix connects the GAFb domain to the catalytic adenylyl cyclase domain of CyaB2. This suggests that the structural changes observed here are signatures of allosteric signal transduction from the GAFb domain to both the GAFa and the adenylyl cyclase domain. The absence of any change in the BRET signal observed with the GAFb sensor construct indicates that the increase in amide exchange following cAMP binding arose from an increase in the entropy of these parts of the GAFb domain, and not as a consequence of a gross change in the relative structure of the protein.

Binding of cGMP, on the other hand, resulted in remarkably less changes in the amide exchange of the GAFb domain, and only some regions in the ligand-binding pocket showed an increase in exchange compared to the unliganded protein (Fig. 4B). A comparison of the exchange profile of the GAFb domain in the presence of cAMP and cGMP clearly showed a number of differences (Fig. 4C), especially in the region containing peptide (305–317) harboring the residue I308. This is in agreement with direct cyclic nucleotide ligand binding data, and confirmed that indeed cAMP and cGMP bind to the GAFb domain in distinct modes. In addition, a lack of alteration in the dynamics of the terminal helices in the presence of cGMP provides a structural basis for the lack of allosteric regulation induced by cGMP binding to the GAFb domain (*Bruder et al., 2005*).

## DISCUSSION

Most cyclic nucleotide binding GAF domains are specific in terms of ligand binding. Efforts have been directed towards understanding the mechanism by which these domains achieve specificity. Mutational and biochemical analysis of GAF domains from other proteins have provided some understanding of the mechanism by which these structurally similar domains discriminate nucleotides (*Linder et al., 2007*; *Schultz, 2009*). We propose

that in addition to the specific interaction of certain residues with the chemical groups present in nucleotides, the hydrophobic interaction provided by residue equivalent to I308 help GAF domains in selecting a specific cyclic nucleotide. Mutations equivalent to I308A in the GAFa domain of PDE5 (*Sopory et al., 2003*) and the GAFb domain of PDE2 (*Wu et al., 2004*) has been shown to reduce cGMP affinity, while affinity for cAMP were reported to be largely unaffected. The GAFa domain of PDE5 and the GAFb domain of PDE2 are known to bind cGMP with high affinity while the affinity of these GAF domains is much less for cAMP. Thus, it appears that the I308 residue in cyclic nucleotide-binding GAF domains dictates the binding of the high affinity ligand. Importantly, in addition, we show here the involvement of T293 in the GAFb domain of CyaB2 in binding cGMP (Figs. 2G and 2C).

The crystal structure of the cAMP-bound GAFb domain showed that cAMP is largely buried, leading to the speculation that the ligand binding pocket is initially present in an open conformation ready to receive the ligand (*Martinez et al., 2005*). Reduction in the dynamics of peptides spanning the ligand-binding pocket (helix $\alpha 4$, helix $\alpha 3$ and sheet $\beta 3$) in the presence of cAMP confirms that the domain 'closes' on ligand binding. A similar mechanism has been proposed for the GAFa domain of PDE5 on binding cGMP (*Wang, Robinson & Ke, 2010*). Interestingly, it appears that the way in which the 'open' GAF domain 'closes' following binding of either the low affinity or high affinity ligand is different. This kind of structural adaptation could not only be necessary to avoid steric hindrance while retaining interactions between the GAF domain and the ligand, but could also play a role in the way the signal of ligand binding to the GAF domain allosterically regulates the C-terminal catalytic domain.

In addition to the mechanism of ligand specificity by a specific residue in the ligand-binding site, comparison of ligand binding to the isolated GAFb domain and the tandem GAF domains revealed a much higher degree of ligand selectivity. This indicates that the isolated GAFb domain and the tandem GAF domains are structurally and biochemically different. Proteins exist in an ensemble of conformations at steady state. The presence of the second GAF domain could influence ligand-binding behavior of the associated GAFb domain by establishing new steady state conformations, allowing concomitant ligand binding specificity, coupled with precise ligand-induced allosteric regulation of these proteins.

## CONCLUSION

Our results provide insights on the basis of nucleotide selectivity and proximal conformational changes that occur following cAMP binding to the GAFb domain of CyA2. They may also allow a molecular understanding of the 'regulated unfolding' that needs to occur during activation of the C-terminal catalytic domains associated with the GAF domains (*Schultz & Natarajan, 2013*), and also provide a foundation for the design of molecules that could modulate GAF domain function and action.

**Abbreviations**

**BRET**     bioluminescence resonance energy transfer;

**cAMP**     adenosine $3'$, $5'$—cyclic monophosphate;

**cGMP**     guanosine $3'$, $5'$—cyclic monophosphate;

**GAF**     cGMP-specific and -regulated cyclic nucleotide phosphodiesterase, Adenylyl cyclase, and *E. coli* transcription factor FhlA;

**HDXMS**     amide hydrogen/deuterium exchange mass spectrometry.

### Funding

Funding has been provided by the Department of Biotechnology and Science and Technology (SSV), and a fellowship from the Council for Scientific and Industrial Research (KHB), Government of India. Support was also provided by the Mechanobiology Institute, National University of Singapore (NUS), Singapore and Waters Center of Innovation Program (to GSA). Personnel exchanges between laboratories was funded by the NUS-India Research Initiative. KHB is currently supported by a Research Fellowship from the Mechanobiology Institute, NUS, Singapore. The funders had no role in study design, data collection and analysis, decision to publish, or preparation of the manuscript.

### Grant Disclosures

The following grant information was disclosed by the authors:
Funding has been provided by the Department of Biotechnology and Science and Technology.
Council for Scientific and Industrial Research, Government of India.
Mechanobiology Institute, National University of Singapore (NUS).
Singapore and Waters Center of Innovation Program.
NUS-India Research Initiative.
Mechanobiology Institute, NUS, Singapore.

### Competing Interests

The authors declare there are no competing interests.

### Author Contributions

- Kabir Hassan Biswas conceived and designed the experiments, performed the experiments, analyzed the data, wrote the paper, prepared figures and/or tables, reviewed drafts of the paper.
- Suguna Badireddy performed the experiments.
- Abinaya Rajendran performed the experiments, analyzed the data.

- Ganesh Srinivasan Anand conceived and designed the experiments, analyzed the data, wrote the paper, prepared figures and/or tables, reviewed drafts of the paper.
- Sandhya S. Visweswariah conceived and designed the experiments, performed the experiments, analyzed the data, wrote the paper, reviewed drafts of the paper.

**Data Deposition**

The following information was supplied regarding the deposition of related data:
FigShare: http://figshare.com/s/80c4370cd85e11e4b16106ec4b8d1f61.

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
