# Peer review of "Cyclic nucleotide binding and structural changes in the isolated GAF domain of Anabaena adenylyl cyclase, CyaB2"

_PeerJ, doi:10.7717/peerj.882_

## Round 0.1 · original submission · Major Revisions

· Academic Editor

Major Revisions

As you will see from the reports, the two reviewers highlight the interest of your findings; however, at the same time they do ask (particularly reviewer #1) that additional functional data be included to fully support the predictions made from the docking. In particular, I would ask you to focus your efforts in the following points:

- address the impact of another GAFb mutant on the cAMP binding (ref#1). However doing NMR HSQC experiments is certainly beyond the scope of this paper and thus is not mandatory.

- split figure 2A for better clarity (1 panel with cAMP, one with cGMP-1 and one with cGMP-2) (ref#2). It is indeed not clear if I308 is clashing (or not) with cAMP.

Thank you for the opportunity to consider your work for publication. I look forward to receiving your revised manuscript.

Reviewer 1 ·

Basic reporting

Biswas et al describe how GAF (GAFa and GAFb) domains of CyA2 are able to recognize cAMP and cGMP. The authors propose that cAMP and cGMP bind to the GAFb domain in distinct modes.
In general, the article provides new data and insight, however it suffers from a few minor and major issues (see below):

major:
- sentence: ‘Amide hydrogen-deuterium exchange mass spectrometry (HDXMS) experiments, however, revealed the structural basis for cAMP-induced allosteric regulation of the GAF20 domains,…’ is a gross exaggeration as ‘structural basis’ normally refers to NMR or crystal structures!

- competition binding curve of cGMP in Figure 1C and D seems incomplete, can the concentration of cGMP be increased to record a complete displacement of 3H-cAMP?

- does the docking take the discussed ‘opening’ and ‘closing’ mechanism into account?

- this reviewers strongly suggest to evaluate at least another mutant that is more important in the recognition of the GAFb domain to confirm proposed model, suggestion: T293 that forms 2 H-bonds with cAMP. If alternative mode (mode 1) is indeed correct, one should see no effect when mutated to alanine.

- should not the alpha4 helix (351-360) also be involved in binding for mode 1 and therefore give a significant difference signal in the HDXMS experiments?

- I wonder if the authors could consider doing NMR HSQC experiments to determine any structural changes and differences between cAMP and and cGMP binding (considering that the size of GAFb is 161 residues which would be well suitable for NMR)

minor:
- Introduction: PDB codes of existing GAF domains would be useful
- Page 28, Legend of Figure 1: ‘assays performed thrice in duplicate’
- Page 9, line 16/17: Protein aggregates was solubilized using urea and was used for antibody generation. Plural!
- Page 13, line 17: with an IC50 OF….
- Keep ‘Mode’ or ‘mode’ consistent throughout paper

Experimental design

as commented above I think NMR HSQC experiments would be better to detect any structural changes upon binding of cAMP/cGMP.
other than that the experimental design is OK

Validity of the findings

the authors need to do another mutants other than the one that is shown, eg. as suggested above - T293 to validate their docking model

Additional comments

NA

·

Basic reporting

Generally fine.

Figure 2A: Can this figure be split into two to make the binding modes clearer? E.g. cGMP-1 and cGMP-2 in one and a corresponding one with cAMP only. The red and magenta (red) molecules/orientations are difficult to discern in the figure. Can a better view be used to show how I308 interacts with cAMP but not to cGMP-1? At the moment the view appears to show I308 clashing with cAMP.

Experimental design

Generally fine.

Can the authors please comment on/confirm whether the BRET-based sensor constructs are in apo state at the experiment? I.e. Can the lack of signal change upon cAMP/cGMP addition or by treatments with forskolin/SNP be attributed to the GAFb is already in a bound state (by cAMP/cGMP or even other molecules)?

Given GAFa plays such an important role in selecting cAMP over cGMP, would it be more appropriate to carry out the HDXMS experiment with a construct that include both GAFa and GAFb?

Validity of the findings

OK

Additional comments

Can the authors please explain in more details why an increase in entropy in parts of the GAFb domain upon cAMP binding would not also result in changes in relative domain positions/orientations?

---

## Round 0.2 · accepted · Accept

· Academic Editor

Accept

I sent this back to one of the original reviewers and took a look myself. We both find your revised paper to be improved and is thus suitable for publication in its current form. Thanks you again for your contribution to PeerJ.

Reviewer 1 ·

Basic reporting

all comments have been addressed and this reviewer recommends acceptance

Experimental design

OK

Validity of the findings

OK

Additional comments

N/A